# Transient Receptor Potential Channels: Important Players in Ocular Pain and Dry Eye Disease

**DOI:** 10.3390/pharmaceutics14091859

**Published:** 2022-09-02

**Authors:** Darine Fakih, Tiffany Migeon, Nathan Moreau, Christophe Baudouin, Annabelle Réaux-Le Goazigo, Stéphane Mélik Parsadaniantz

**Affiliations:** 1Laboratoires Théa, 12 Rue Louis Blériot, CEDEX 2, 63017 Clermont-Ferrand, France; 2CNRS, INSERM, Institut de la Vision, Sorbonne Université, 17 Rue Moreau, 75012 Paris, France; 3CHNO des Quinze-Vingts, INSERM-DGOS CIC 1423, 28 Rue de Charenton, 75012 Paris, France; 4Department of Ophtalmology, Ambroise Paré Hospital, AP-HP, 9 Avenue Charles de Gaulle, 92100 Boulogne-Billancourt, France

**Keywords:** ocular pain, dry eye, TRPV1, TRPM8, topical treatment

## Abstract

Dry eye disease (DED) is a multifactorial disorder in which the eyes respond to minor stimuli with abnormal sensations, such as dryness, blurring, foreign body sensation, discomfort, irritation, and pain. Corneal pain, as one of DED’s main symptoms, has gained recognition due to its increasing prevalence, morbidity, and the resulting social burden. The cornea is the most innervated tissue in the body, and the maintenance of corneal integrity relies on a rich density of nociceptors, such as polymodal nociceptor neurons, cold thermoreceptor neurons, and mechano-nociceptor neurons. Their sensory responses to different stimulating forces are linked to the specific expression of transient receptor potential (TRP) channels. TRP channels are a group of unique ion channels that play important roles as cellular sensors for various stimuli. These channels are nonselective cation channels with variable Ca^2+^ selectivity. TRP homologs are a superfamily of 28 different members that are subdivided into 7 different subfamilies based on differences in sequence homology. Many of these subtypes are expressed in the eye on both neuronal and non-neuronal cells, where they affect various stress-induced regulatory responses essential for normal vision maintenance. This article reviews the current knowledge about the expression, function, and regulation of TRPs in ocular surface tissues. We also describe their implication in DED and ocular pain. These findings contribute to evidence suggesting that drug-targeting TRP channels may be of therapeutic benefit in the clinical setting of ocular pain.

## 1. Introduction

Dryness and pain are common devastating symptoms of DED, affecting the quality of life of 10% of the population worldwide. The most recent definition of DED was published in 2017 by the international report DEWS II. DED is “a multifactorial disease of the ocular surface characterized by a loss of homeostasis of the tear film, and accompanied by ocular symptoms, in which the instability and hyperosmolarity of the tear film, inflammation and damage to the ocular surface and neurosensory abnormalities play an etiological role” [1]. Based on the definition of TFOS DEWS II, DED is classified into two types: EDE results from excessive evaporation of the tear film with normal tear function, and ADDE results from reduced tear secretion with regular evaporation of the tear film [2] and, therefore, mixed DED, which is the assembly of the two classes. Classification is not the only complex issue of DED; another is the definition of the precise underlying pathophysiology. In 2016, Baudouin et al. proposed several interconnected etiologies as entry points into the vicious circle of DED: tear film instability, tear hyperosmolarity, apoptosis, and inflammation [3,4]. Thus, tear hyperosmolarity stimulates a cascade of events in the epithelial cells of the ocular surface, involving impaired MAP kinases and NF-κB signaling pathways, the generation of inflammatory cytokines, and the induction of oxidative stress [5,6]. These lead to a reduction in mucin expression, the death of surface epithelial cells, and a loss of goblet cells, which, in turn, compromise the wetting of the ocular surface. Finally, hyperosmolarity of the ocular surface is amplified, which completes the vicious circle of dry eye and establishes the mechanism that perpetuates the disease. However, tear hyperosmolarity is not necessarily the only starting point for DED [4]. The management of dry eye is complicated due to its multifactorial etiology. Current therapies include treatments for lacrimal insufficiency and eyelid abnormalities, anti-inflammatory drugs, surgical approaches, certain dietary modifications, and environmental considerations.

The International Association for the Study of Pain (IASP) defines pain as “an unpleasant sensory and emotional experience associated with, or resembling that associated with, actual or potential tissue damage”. Pain can be of nociceptive or neuropathic origin. Nociceptive pain is caused by a damage to tissues, leading to the activation of nociceptors, and comprises thermal pain, chemical pain, and mechanical pain. Neuropathic pain results from damage to the nervous system. Typical characteristics of neuropathic pain contain burning, local sensory deficit; hyperalgesia (amplified pain response to noxious stimuli); allodynia (pain sensation by innocuous stimulus, such as light touch); and ectopic discharges [7].

Several studies describing the symptoms of patients with DED report corneal neuropathic pain in patients without apparent corneal damage or after resolution of their corneal damage. These symptoms comprise hypersensitivity to light (photophobia), wind, and heat or cold; spontaneous eye burning; and pain to pressure and light touch or during the application of Schirmer’s test strips [7,8]. In dry eye, reduced tear secretion leads to inflammation and causes sensitization of polymodal and mechano-nociceptor nerve endings and cold thermoreceptors. Molecular, structural, and functional perpetuations in trigeminal sensory pathways ultimately lead to neuropathic pain of the ocular surface.

This article reviews the current knowledge regarding the expression, function, and regulation of TRPs in the anterior segment of the eye by focusing on their implication in DED and ocular pain. These findings contribute to evidence suggesting that drugs targeting TRP channels may provide therapeutic benefits in the clinical setting of ocular pain.

## 2. Innervation of the Cornea

The cornea is a transparent avascular tissue and the most innervated tissue in the body [9,10,11]. It is only innervated by nociceptive Aδ and C fibers, which terminate as free nerve endings morphologically similar to those observed in the skin [12]. C fibers are between 0.3 and 1.5 µm in thickness. Their conduction velocity is about between 0.4 and 2 m/s; C fibers represent 80% of corneal fibers and are responsible for “second” or slow pain [13,14]. Aδ fibers are about 1–5 µm in thickness; their conduction velocity is between 4 and 30 m/s. Aδ fibers represent 20% of corneal fibers and are more implicated in acute pain generation because of their capability for fast transmission of nociceptive messages [13,14].

Corneal innervation is provided by peripheral axons of neurons located in the dorsomedial portion of the ophthalmic region of the trigeminal ganglion (TG) [9,15,16]. The TG is the main origin of cerebrovascular sensory nerve fibers, and it contains sensory neurons from the ophthalmic (V_1_), maxillary (V_2_), and mandibular (V_3_) divisions of the trigeminal nerve [15]. Different studies estimate that 50 to 450 trigeminal neurons innervate the cornea, and these represent only 1% to 3% of the total population within the TG [9,17]. Furthermore, corneal neurons project their central axons to the trigeminal brainstem sensory complex (TBSC), terminating in two distinct regions of the TBSC: the trigeminal subnucleus interpolaris/caudalis (Vi/Vc) transition region and the subnucleus caudalis/upper cervical cord (Vc/C1) junction region [16,18]. Then, second-order neurons synapse with third-order neurons in the thalamus. The activated neurons at the thalamus, periductal gray, and amygdala levels send top-down inhibitory signals (descending pathways) to active synapses in the TBSC, thus modulating the activity of ascending excitatory nociceptive pathways [19]. Thus, a loss of inhibitory neurotransmitters such as gamma aminobutyric acid (GABA) in trigeminal sensory pathways may lead to acute and chronic pain conditions.

The mechanisms responsible for maintaining corneal integrity include a uniquely rich density of nerve terminals called nociceptors estimated to be up to 40-fold greater than that of dental [20] and 100-fold greater than that of skin. The cornea is protected with the highest density of nociceptor nerve endings, estimated at 16,000 nerve terminations/mm^3^ [21]. The distal terminals of corneal nociceptors contain sodium ion channels capable of generating action potentials (AP), which explains that the cornea is the most powerful pain generator in the human body [22].

### Corneal Nociceptors and Cold Thermoreceptors

Different types of receptors (polymodal nociceptors, thermoreceptors, and mechano-nociceptors) coexist on a single corneal nerve ending [23]. Their sensory responses to various stimulating forces are linked to a specific expression of ion channels [9,23].

Polymodal nociceptors represent the most represented corneal nociceptors (41%) [13]. They are activated by mechanical stimulations (characterized by a lower mechanical threshold than mechano-nociceptors) [24], heat (starting to fire at 37 °C) [25], chemical irritants, and acidic pH [25,26,27,28]. Furthermore, corneal polymodal nociceptors can be activated by inflammatory mediators and endogenous mediators produced by damaged cells or inflammatory cells [29]. Most of them are of the slow-conducting C nerve fibers. However, some of these fibers belong to the thin Aδ myelinated group [13].

Corneal cold thermoreceptors discharge continuously for temperatures between 34 and 35 °C. Their activity is influenced by moderate temperature variations (cooling and heating) [27,30]. Cold thermoreceptors are located on Aδ and C nerve fibers and classified into two different subtypes: 34% of cold sensitive thermoreceptors have a high-background, low-threshold (HB-LT) activity, whereas the other 15% have a low-background, high-threshold (LB-HT) activity. HB-LT cold thermoreceptors detect 0.5 °C temperature changes on the corneal surface, while LB-HTs’ firing activity increases in response to >4 °C changes in temperature [31,32,33] (Figure 1).

Finally, the firing activity of mechano-nociceptors (which are expressed in Aδ nerve fibers and represent 10% of total nociceptors) increases only secondary to mechanical forces but remains silent in the absence of stimulation [26,35]. Furthermore, the activation of corneal mechanonociceptors results in instant and severe sensation of pain [9].

## 3. The TRP Channels and Their Implication in the Physiology and Pathophysiology of the Anterior Segment of the Eye

In 1969, the first transient receptor potential (TRP) channel was discovered in a mutant strain of *Drosophila melanogaster*. Deficiency of the TRP gene resulted in damage to the Drosophila’s visual system [36]. Later, in 1992, Roger Hardie and Baruch Minke showed that the TRP gene is crucial for a light-activated Ca^2+^ channel in the fly’s photoreceptors [37]. Initially, TRP represented a light-sensitive channel, and this was supported by the cloning of the second light-sensitive channel, TRPL. Then it was shown that TRPL plays a crucial role in maintaining a sustained response during prolonged illuminations [38]. Since then, different TRP homologues have been identified.

TRP channels function as nonselective cation-permeable channels and, when activated, depolarize the cell, leading to subsequent voltage-dependent ion channel activation and changes in intracellular Ca^2+^ concentrations. Thus, this may explain the implication of TRP mutations in diverse diseases, such as neurodegenerative diseases, skeletal dysplasia, kidney diseases, and pain [39,40].

TRP channels are composed of six transmembrane domains with four subunits surrounding a centrally located ion permeation pore with intracellular N and C terminals. Their classification is based on sequence similarities and not on their common functional features. Regarding this classification, members of the same family may be functionally different, and members from different families may share common features. TRP homologs in mammalian species represent 28 channels, grouped into several families: TRPC (canonical), TRPM (melastatin), TRPN (NOMPC-like, found only in invertebrates and fish), TRPV (vanilloid), TRPA (ankyrin), TRPML (mucolipin), and TRPP (polycystin) [41,42,43,44] (Figure 2).

TRP channels are a group of unique ion channels that play role as cellular sensors for varied stimuli, such as changes in pH, pungent peppers, wasabi, mustard, and menthol, as well as thermal, mechanical, osmotic, and actinic (radiation) signals [45,46]. TRP channels can be activated directly (by thermal and mechanical stimuli) by an exogenous ligand, by a lipidic endogenous ligand, or by receptor activation (G protein–coupled receptors (GPCR) and receptor tyrosine kinases) [46] (See Table 1).

TRP channels located on the corneal nerves lead to an influx of cations, which shifts the cell membrane potential to depolarization. This in turn generates AP, which actively propagates centripetally along small-caliber Aδ and C primary afferent axons and synapse in the TBSC [19]. In the sensory nervous system, GPCR and TRP channels are coexpressed by peptidergic neurons that detect nociceptive, irritating, and inflammatory stimuli. Thus, it has been established that the GPCR plays a key role in the amplification of pain and neurogenic inflammation by sensibilization mechanisms of TRP channels, leading to their activation [47].

### 3.1. TRPC Family

TRPC is a nonselective Ca^2+^-permeable cation channel. The TRPC family contains seven members who can be grouped, based on sequence homology, into four subfamilies: TRPC1, TRPC2, TRPC4/5, and TRPC3/6/7. Due to the highest protein sequence similarity to the TRP channel of *Drosophila melanogaster*, the first human homologue to be cloned was TRPC1, in 1995 [48,49].

The two main activation mechanisms of each member of the TRPC family are extensively described in the review by Chen et al. [50]. Recently, Bon et al. reviewed the pharmacology of TRPC channels and the potential effects of its modulators in the treatment of cardiovascular disease [51]. The most described TRPC channel activity modulation mechanisms are mediated by calmodulin (CaM). Indeed, all TRPC channels have CaM-binding sites located on their cytoplasmic domains [52]. It has been shown that the interaction of Cam-Ca^2+^ with the CaM-binding sites of TRPC channels leads to the inhibition of its activity, whereas the interaction of IP3 receptors with CaM-binding sites leads to activation [53]. These results have been recently confirmed, revealing new pharmacological tools to manipulate this therapeutic target [54].

Interestingly, it has been recently demonstrated that TRPC channels are expressed by neurons. Therefore, they could be involved in the transduction of mechanosensitive stimuli. Indeed, in vitro dorsal root ganglion neurons have a decreased response to stretch activation when TRPC1 channels are downregulated [55]. Furthermore, TRPC1 knock-out mice present lower behavioral responses to innocuous mechanical stimuli, but these mice present no change in their behavioral responses to noxious mechanical stimuli [56]. Another study concludes that there are some interaction and cooperative mechanisms between TRPV4, TRPC1, and TRPC6, which results in the development of mechanical hyperalgesia [57]. Finally, using double TRPC3/TRPC6 knock-out models, it has been demonstrated that these channels are required for normal mechano-transduction [58]. Although it was shown that the TRPC channels are involved in mechanosensitive transduction, it has yet to be understood how a mechanical stimulus can activate these channels.

A study reported TRPC4 expression in bovine corneal endothelial cells [59]. Thus, it was hypothesized that TRPC4 could inhibit receptor-mediated calcium entry in these cells. In addition, Yang et al. demonstrated that TRPC4 is also expressed in human corneal epithelial cells (HCECs) [60]. Thus, they evidenced that TRPC4 can be involved in corneal epithelial cell proliferation. Indeed, TRPC4 is essential for optimal mitogenic response to the epidermal growth factor.

### 3.2. TRPP Family

The transient receptor potential polycystin (TRPP) channel family contains three different members: TRPP2, TRPP3, and TRPP5. The TRPP2 channel is also named polycystin-2 (PC-2) and is encoded by the polycystic kidney disease 2 (PKD2) gene. This channel is largely described because of its involvement in autosomal dominant polycystic kidney disease (ADPKD) [61]. This channel is a nonselective Ca^2+^ permeable cation channel. TRPP2 is composed of four domains, and each domain is constituted by six transmembrane segments [61]. The C-terminal domain of TRPP2 is an EF-hand domain, a flexible linker, and an oligomeric coiled coil domain [62]. The EF-hand domain corresponds to helix–loop–helix motifs that have Ca^2+^ binding sites. Ca^2+^ ions are responsible for either TRPP2 activation or inhibition. As TRPP2 is especially localized on the endoplasmic reticulum membrane, its activation leads to intracellular Ca^2+^ release [62]. Regarding the two other members of the TRPP subfamily, they have been identified via their high homology to TRPP2. They are also nonselective Ca^2+^ permeable cation channels, and they have a structure like TRPP2.

Regarding the ocular expression of TRPP2, its immunostaining was observed in the basal cell layer of the corneal epithelium [63,64]. TRPP2 is also detected in the acinar tear cells of mice [65]. Like in other tissues, TRPP2 in the lacrimal gland is especially expressed on the membranes of the endoplasmic reticulum and has an important role in Ca^2+^ signaling. Thus, these studies suggest that targeting TRPP2 could be an innovative strategy in eye diseases affecting tear production [65].

### 3.3. TRPA Family

The only representative of this family in mammals, TRPA1 is a nonselective cationic TRP channel that can be activated by many natural and synthetic irritants, such isothiocyanate or thiosulfinate compounds, methyl salicylate, ginger, carvacrol, formalin, natural fungal deterrents such as isovelleral, unsaturated aldehydes such as acrolein, isocyanates, cinnaldehyde, oxidizing agents [66], and hydrogen peroxide [67,68]. Additionally, TRPA1 plays an essential role in corneal neovascularization. During the corneal wound healing process, the transactivation of TRPA1 by vascular endothelial growth factor (VEGF) controls neovascularization and macrophage infiltration [69].

TRPA1 activation excites sensory nerve fibers and produces acute pain and neurogenic inflammation via the peripheral release of neuropeptides (SP and CGRP) and purines. TRPA1 is also activated by temperatures below 17 °C. In addition, TRPA1 is considered an essential component of mechanically gated transduction channels in auditory hair cells [70]; at the spinal level, TRPA1 is also involved in mechanisms related to mechanical hypersensitivity [71]. TRPA1 is expressed peripherally in the cornea, in 35% of the TG neurons, and centrally in the Vc nucleus, and its expression in trigeminal nonpeptidergic neurons is regulated by the cornea. Canner et al. cocultured corneas and the ophthalmic part of the TG in vitro to test interactions between nerve and cornea. They showed that TRPA1 expression was increased in the ganglion when cocultured with cornea, compared with isolated ganglia cultures. These findings highlight that corneal-derived factors increase TRPA1 expression in trigeminal neurons [72].

In immunohistochemical experiments on mouse and monkey corneas, Schecterson et al. demonstrated that TRPA1 is colocalized with secretogranin 3 in intracellular vesicular structures located in axons with large dense vesicles [73]. In addition, TRPA1 is involved in acute ocular pain, as well as in painful sensations during allergic keratoconjunctivitis or other ophthalmic conditions, as blocking or reducing channel expression ameliorates ocular pain. For review, see [74]. Furthermore, TRPA1 is upregulated by proinflammatory molecules found in tears of DED patients, and its activation is linked to chronic itch, a common symptom in DED patients [75]. Furthermore, Katagiri et al. were interested in studying TRPA1 involvement in a model of DED. They induced DED by the excision of the extraorbital lacrimal gland in rats and evaluated the effects of the topical administration of mustard oil (0.02% and 0.20%), a TRPA1 agonist. They observed that such TRPA1 agonist increased eyeblink and forelimb eye wiping behavior in DED and sham rats, increased c-Fos immunoreactivity (neuronal activation marker) in the Vi/Vc transition, in the midportions of Vc and in the trigeminal (Vc/C1) region of DED rats. However, TRPA1 protein levels from ipsilateral and contralateral TG of DED rats were the same. To sum up, these findings illustrate that TRPA1 plays an important role in the sensitization of ocular-responsive trigeminal brainstem neurons in DED [75].

Otherwise, Hirata et al. evaluated the responses of corneal cold-sensitive neurons to a series of wet and dry stimuli before and after short ocular application of the TRPA1 antagonist 20 μm HC030031. These results showed that HC030031 did not influence the responses to corneal drying [76]. In addition, Parra et al. applied the TRPA1 agonist allyl isothiocyanate (AITC, 100 μm) in cold-sensitive nerve terminals of the cornea and recorded nerve terminal impulse activity in mouse eyes in vitro. They showed that AITC did not affect the ongoing or cold- and menthol-evoked activity in cold neurons. These results indicate that TRPA1 is not a molecular determinant for corneal cold sensitivity [33].

Conversely, Acosta et al. were interested in investigating the sensations of irritation, discomfort, and itch accompanying the allergic eye response [77]. For this purpose, they measured the blinking and tearing rate and recorded the firing of corneo-conjunctival sensory nerve fibers of the guinea pig after an ovalbumin allergic conjunctivitis. Treatment with capsazepine (5 mm) and HC-030031 (100 μm) reversed the increased blinking. Only capsazepine reduced tearing rate increase and sensitization of polymodal nociceptors’ response to CO_2_. Furthermore, it prevented the decrease in cold thermoreceptor activity caused by the allergic challenge. These results highlight that TRPV1 is more involved than TRPA1 in the mechanisms underlying ocular allergy [77].

The release in the atmosphere of the pesticide composed of methyl isocyanate (a TRPA1 agonist) in Bhopal, India, caused the worst industrial accident in history. Over 500,000 people were exposed to methyl isocyanate gas and caused the death of 16,000 to 30,000 people. Bessac et al. studied the noxious effects of isocyanates on ocular irritation and pain. They showed that in TRPA1^−/−^ mice, isocyanate- and tear gas-induced nocifensive behaviors were decreased following both ocular and cutaneous exposures, which highlights that methyl isocyanate targets TRPA1 and causes chemical irritation via this channel [78]. Moreover, TRPA1 was evaluated in a UV-induced keratitis in guinea pigs. Acosta et al. observed a decrease in mechanical threshold in vivo after TRPA1 agonist instillation in UV-irradiated animals. It is known that the increased expression of TRPA1 during inflammation is associated with mechanical hyperalgesia. Thus, TRPA1 activation could explain the increase in nocifensive behaviors in UV-induced keratitis [79].

### 3.4. TRPM Family

The TRPM family is divided into four groups: TRPM1/3, TRPM2/8, TRPM4/5, and TRPM6/7. TRPM channels are characterized by their highly varying permeability to Ca^2+^ and Mg^2+^, from Ca^2+^-impermeable (TRPM4/5) to highly Ca^2+^- and Mg^2+^-permeable (TRPM6/7) channels. In contrast with the TRPV family, the TRPM sequence does not contain ankyrin repeats. The TRP box is located at the C-terminal domain in both TRPM and TRPV families [80].

#### 3.4.1. TRPM8

TRPM8 is a cold- and menthol-sensitive nonselective cation channel [81] expressed by the subpopulation of sensory trigeminal neurons [82,83,84] but also by corneal keratocytes [85]. The genetic deletion of TRPM8 revealed that TRPM8^−/−^ mice are deficient in unpleasant cold sensitivity, confirming the implication of this channel in cold-pain sensations [86,87,88].

Pina et al. were interested in evaluating thermal and chemical sensitivity, excitability, and TRPM8 functional expression in mice cold thermoreceptors following damage of corneal nerve fibers [84]. They observed that the surgical injury of corneal peripheral nerves increased the percentage of corneal cold-sensitive neurons, the cold- and menthol-evoked intracellular [Ca^2+^] concentration, and the ongoing firing activity and menthol sensitivity. These results highlight the implication of cold nociceptors in the sensation of discomfort, pain after photorefractive surgery, and DED. Moreover, Parra and al. showed that the deletion of TRPM8 in TRPM8^−/−^ mice abolishes cold responsiveness and diminishes basal tearing without disturbing nociceptor-mediated irritative tearing [33]. Another study on aged mice (24 months) showed that aging impairs the activity of high-threshold cold thermoreceptors. This observation highlights that the hyperactivity of the cold thermoreceptor on the cornea may play a role in the high incidence of DED in aged people [82]. In addition, DED patients showed greater sensitivity compared with healthy subjects [89]. A recent study showed that TRPM8 is a pharmacological target of tacrolimus (FK506), a macrolide immunosuppressant with several clinical uses, including the treatment of organ rejection following transplants, treatment of atopic dermatitis, and DED.

In the literature, there are two different opinions about the use of a TRPM8 agonist vs. a TRPM8 antagonist in DED. Some studies consider that TRPM8 agonists can be an effective treatment for DED, while others suggest that TRPM8 antagonists could relieve DED [90]. In this regard, pharmacological studies have demonstrated that TRPM8 ligands or inhibitors/blockers attenuate pain sensation in numerous somatic pain models. TRPM8 antagonists have been shown to efficiently alleviate acute and chronic pain [91,92,93,94], whereas the TRPM8 agonist may present significant antiallodynic activity through an excessive activation of TRPM8, leading to its downregulation and/or desensitization [95]. Recently, C. Izquierdo et al. reviewed in depth the different structural studies of the TRPM8 channel and its pharmacological modulation by specific agonists/antagonists [96].

##### TRPM8 Agonists to Alleviate DED and Ocular Pain

In animal models of DED, topical menthol (50 μm) increased the cooling sensation in [31]. Furthermore, Hegarty et al. evaluated evoked ocular sensory responses in rats after lacrimal gland denervation and hypothesized that decreased TRPM8 function in corneal sensory nerves may play a role in ocular hypoalgesia [97]. Additionally, Robbins et al. indicated that low concentrations of menthol can increase lacrimation in mice via TRPM8 without causing nocifensive behaviors; however, at high concentrations, it induces lacrimation and nocifensive behaviors [98]. In a rat model of DED, topical menthol (50 μm) did not enhance eye wiping behavior in DED animals. DED rats did not have greater orbicularis oculi muscle (OOemg) activity after menthol compared with sham.

In a study of Yang et al. [99], cooling relieved postcataract surgery pain by decreasing corneal and conjunctival nociceptive sensibilities. In addition, Kurose et al. showed that menthol-induced desensitization of corneal cool cells may reduce tearing, a harmful effect in individuals with DED. Nevertheless, the use of TRPM8 agonists, such as menthol (contained in some over-the-counter eye drops) is contradicted and have limited value in ocular studies in humans. Indeed, menthol vapors irritate the eye, cause discomfort, and may have damaging effects in DE sufferers [100]. Yan et al. evaluated a soluble TRPM8 receptor agonist called cryosim-3 (C3, 1-diisopropylphosphorylnonane), which selectively activates TRPM8 in DED patients. C3 generated a cooling effect in less than 5 min (lasting 46 min) with an increase in tear secretion for 60 min. No complaints of irritation or pain were reported by any subject [101]. In addition, Jeong Yoon et al. investigated in a clinical study the effect of a topical TRPM8 agonist, C3, on relieving DE-associated neuropathic ocular pain. After topical application of C3 to the eyelid, four times/day for 1 month, they demonstrated that Ocular Pain Assessment Survey (OPAS) scores of eye pain intensity, quality of life, and Schirmer test of DED patients were improved for 1 month [102]. Additionally, some patents reported the use of TRPM8 agonists to relieve ocular pain and dry eye.

Cinnamomum camphora chvar (borneol) is a natural compound widely used in ophthalmic preparations in China. Thus, Chen et al. introduced borneol as a treatment for DED [103]. They observed that topical borneol significantly increased tear production in guinea pigs without inducing nociceptive responses at 25 °C, but failed to induce tear secretion at 35 °C. Besides, this agent did not affect the viability of human corneal epithelial cells. However, to date, the therapeutic use of borneol in DED is only authorized in China.

##### TRPM8 Antagonists to Alleviate DED and Ocular Pain

However, the selective antagonism of TRPM8 (AMTB) reduced hypertonic saline evoked orbicularis oculi muscle activity (OOemg) activity [104]. Furthermore, Hirata et al. showed that the TRPM8 antagonist BCTC (20 μm) decreased the drying of the cornea by ~45%–80% but could not completely block them. The authors reasoned that the stimulus of corneal dryness derives partly from TRPM8 channels, but also that non-TRPM8 channels contribute significantly to the dry responses and to basal tearing. Moreover, they theorized that TRPM8 activation by cooling in corneal sensory afferents increases the sensation of ocular coolness (137). Besides, Fakih et al. showed the effectiveness of repeated instillations of the TRPM8 antagonist M8-B in a mouse model of severe DED induced by the excision of extraorbital lacrimal and Harderian glands. DED mice developed cold allodynia consistent with higher TRPM8 mRNA expression in TG (Figure 3 and Figure 4). M8-B (20 μm) was topically administered twice a day from day 7 until day 21 after surgery (Figure 4). Thus, chronic M8-B instillations markedly reversed both the corneal mechanical allodynia and spontaneous ocular pain commonly associated with persistent DED. M8-B instillations also diminished the sustained spontaneous and cold-evoked ciliary nerve activities observed in DED mice as well as inflammation in the cornea and TG [105].

### 3.5. TRPV Family

#### 3.5.1. TRPV2, TRPV3, TRPV4, and TRPV6

TRPV2 is the second member of TRP channels. Although TRPV2 is a nonspecific cation channel, it is more permeable to calcium ions; it is activated by a very high-threshold temperature (>52 °C), several synthetic cannabinoids such as cannabidiol, swelling, and 2-aminoethoxydiphenyl borat (2-APB) [67,106]. TRPV2 is insensitive to capsaicin [107]. It is not activated by vanilloids, protons, and moderate thermal stimuli [74]. Indeed, temperatures activating TRPV2 are more intense than those activating TRPV1 [108]. It is widely expressed in mechano- and thermos-responsive neurons (TG) and non-neuronal cells (immune cells), and it has been identified in the mouse retina and the human conjunctiva [42]. TRPV2^−/−^ mice show normal thermal and mechanical nociceptive behaviors [109]. Conversely, Shimosato et al. showed that TRPV2 plays a role in the peripheral sensitization during inflammation and is responsible for pain hypersensitivity caused by noxious high-temperature stimuli [110].

TRPV3 was first cloned in 2002 and shares 40–50% homology with TRPV1. TRPV3’s implication in the peripheral nervous system is debated, as its expression in sensory neurons is extremely low in rodents. However, it is expressed in non-neural tissues and cell populations, such as the epidermis, keratinocytes, intestinal epithelial cells, and vascular endothelial cells [111]. TRPV3 is activated at 32–39 °C and by natural compounds, such as camphor and several herbs [111]. It is also activated by chemical molecules, such as spice extracts (such as camphor and carvacrol), synthetic agents (2-APB), and the endogenous ligand farnesyl pyrophosphate (FPP). TRPV3 sensitization occurs by repeated exposure to heat or chemical agonists or the activation of Gq-coupled GPCRs [112]. TRPV3 has many physiological and pathophysiological functions. Yamada et al. showed that TRPV3 is expressed in corneal epithelial cells of humans and mice and that TRPV3 activation improves the cellular viability and increases the wound healing rate of HCE-T cells [113].

By an immunohistochemical method, Ai Izutani-Kitano et al. compared the expression pattern of TRPV2 and TRPV3 in healthy and diseased conjunctival epithelium. Thus, they demonstrated that expression profiles of TRPV2 and TRPV3 are altered in ocular surface neoplasm, which could be a potential diagnostic marker for ocular surface epithelial disorder [114].

TRPV4 is a cation channel activated by hypotonicity-induced cell swelling, moderate heat (>24 °C to 27−34 °C), flow, endogenous ligands such as endocannabinoids, and metabolites of arachidonic acid, as well as synthetic agonists, such as GSK1016790A (GSK101) [115]. TRPV4 participates in numerous cellular functions, such as mechanosensation and thermoregulation. Besides, TRPV4 contributes to chondrocyte and osteoclast differentiation in arterial smooth muscle and endothelial breast cancer cell migration and in UVB radiation-induced pain and skin injury [116,117]. Additionally, TRPV4 is expressed in anterior segment tissues, such as corneal epithelium, ciliary body, and the lens. TRPV4 channels are essential for maintaining normal ocular physiology [115].

TRPV4 has been well documented in human epidermal keratinocytes and is known to strengthen tight junction (TJ) barrier function in these cells. Thus, Okada et al. studied the implication of this channel in corneal epithelial cell differentiation. They showed that TRPV4 was essential for the correct establishment of TJs in corneal epithelia. Moreover, fluid secretion by the ciliary body is an essential function in vertebrate vision, which provides nutritive support to the cornea and lens and maintains intraocular pressure homeostasis. Jo et al. showed that TRPV4 controls cell volume, lipid, and calcium signals in nonpigmented and pigmented epithelial cells of the mouse ciliary body, which highlights that TRPV4 can be a promising drug target for the treatment of glaucoma [118]. Besides, Pan et al. reported a functional expression of TRPV4 in HCECs and that its activation by exposure to a hypotonic challenge is necessary for inducing regulatory cell volume decrease behavior [119].

In a murine model of corneal alkali burn, loss of TRPV4 gene function or intraperitoneal injection of an TRPV4 antagonist reduced alkali-burn-induced fibrotic and inflammatory responses in the mouse cornea. These findings highlight TRPV4’s implication in corneal wound healing [120]. Furthermore, Okada et al. showed that the sensory nerve TRPV4 is essential to maintain the stemness of peripheral/limbal basal cells and is one of the key mechanisms of corneal epithelium homeostasis in a mouse model of neurotrophic keratopathy obtained by coagulating of the first branch of the trigeminal nerve [121]. In addition, Shahidullah et al. showed that TRPV4 in porcine lens epithelium regulates ATP release, which stimulates the P2Y receptor and activates the Na^+^/K^+^ ATPase pump to maintain lens transparency [122].

Corneal epithelial surface microvilli help the tear film to bond to the superficial corneal epithelial cells. Tang et al. were interested in evaluating whether sleep deprivation (for 20 h per day for 5 or 10 days) induces DED through peroxisome proliferator-activated receptor alpha (PPARα) expression alteration in mice. They showed that sleep deprivation–induced corneal epithelial lipid accumulation and microvilli morphologic change decreased tear production and decreased in PPARα and TRPV6 expression levels. Moreover, the application of fenofibrate (a PPARα agonist) increased PPARα and TRPV6 gene and protein expression levels and restored microvilli morphology in corneal epithelial cells. The conclusion of this work was that sleep deprivation caused DED via changes in superficial corneal epithelial cell microvilli morphology, secondary to the downregulation of PPARα and TRPV6 expression [123].

#### 3.5.2. TRPV1

TRPV1 was first cloned by Caterina and colleagues in 1997 [124]. It is a molecular marker for polymodal nociceptors and is also known as “Vanilloid receptor 1” and “capsaicin receptor”. TRPV1 is a nonselective cation channel with a preference for calcium, characterized by six transmembrane segments and structural resemblance with potassium channels [125]. TRPV1 is sensitive to heat (≥42 °C), protons (low pH), and hyperosmolarity [124,126,127]. It can be stimulated by a large array of endogenous ligands: extracellular acidic environment [124,128,129], intracellular basic environment [130], lipid metabolites such as endovanilloids and endocannabinoids [131,132], fatty acid derivatives, oxygenated eicosatetraenoic acids and lysophosphatidic acid [131,133,134], adenosine triphosphate (ATP), adenosine and polyamine [135], exogenous ligands from dietary products (capsaicin, piperine, eugenol, and gingerol) [136], plant toxins (resiniferatoxin from the cactus Euphorbia resinifera) [137], and animals toxins (spiders and jellyfish) [138,139].

TRPV1 sensitization happens in two different ways: indirectly by the activation of GPCR or tyrosine kinase receptor pathways or directly by some agents [140]. Furthermore, inflammatory mediators (i.e., bradykinin, prostaglandin E2, extracellular ATP, glutamate, and nerve growth factor (NGF)) indirectly sensitize TRPV1 [141].

##### Effect of TRPV1 on Corneal Epithelial and Endothelial Cells

TRPV1 expression was demonstrated in corneal epithelial and endothelial cells [142,143,144] but also in keratocytes [85,145]. Thus, Pan et al. showed that hyperosmotic stress and IL-6 and IL-8 releases were decreased by the incubation of HCECs with capsazepine, a TRPV1 antagonist [146].

The maintenance of corneal transparency is reliant on the stroma remaining avascular nonfibrotic and free of inflammation. Stromal neovascularization is a biological reaction observed in the injured cornea. Once the cornea is injured, epithelial cells and keratocytes increase their expression of wound-healing-promoting cytokines. Transforming growth factor β1 (TGFβ1) is one of the most effective cytokines that takes part in the healing process. Tomoyose et al., using a TRPV1 KO mice model, proved that lack of TRPV1 inhibited neovascularization in corneal stroma following cauterization and suppressed vascular endothelial growth factor (VEGF) and TGFβ1 mRNA expression in a mouse cornea [147]. Moreover, Nidegawa-Saitoh et al. demonstrated on TRPV1^−/−^ mice the implication of TRPV1 in corneal healing [148]. Likewise, corneal alkali burn is an important ophthalmological problem and may cause visual impairment by inducing tissular inflammation. Okada et al. used TRPV1^−/−^ and TRPV1^+/+^ mice and showed that the inhibition of TRPV1 activation after an alkali burn markedly reduces corneal fibrosis inflammation and opacification [149].

Yang et al. showed a colocalization between CB1 and TRPV1 in the intact mouse corneal epithelium and in HCECs. Moreover, they established that TRPV1 and cannabinoid receptor 1 (CB1) (a G-protein-coupled receptor, activated by endocannabinoids and synthetic agonists such as WIN55,212-2 (WIN) functionally interact in HCECs. Thus, CB1 and TRPV1 activation induces an increase in HCEC proliferation and migration through EGFR transactivation, leading to global MAPK and Akt/PI-3K pathway stimulation. Moreover, they demonstrated that TRPV1 mediates the increase in proinflammatory cytokine (IL-6 and IL-8) release through both EGFR-dependent and EGFR-independent signaling pathways [150]. Furthermore, whole-cell patch clamp analysis of HCECs showed that WIN suppressed capsaicin-induced cation channel currents in HCECs. Thus, CB1 activation contributes to TRPV1 dephosphorylation, leading to TRPV1 desensitization [151].

##### TRV1 and DED

As we saw, dry eye is a multifactorial disease of the tears and ocular surface often accompanied by tear film hyperosmolarity, inflammation of corneal and conjunctival epithelial cells, and decrease in conjunctival goblet cells and mucin production. Thus, TRPV1 seems to play a key role in mediating DED symptoms induced by tear hyperosmolarity. Indeed, chronic tear deficiency enhances the excitability of corneal cold-sensitive nerves through TRPV1-mediated response in the corneal LB-HT cold thermoreceptors and cold-insensitive polymodal nociceptors [152]. Moreover, in HCECs exposed to hyperosmotic stress, it was established that capsazepine (TRPV1 activation inhibitor) can blocked the rise of proinflammatory mediators, EGFR transactivation, MAPK, and NF-κB activation [146].

To date, osmoprotective agents are used in the clinic as therapeutic approaches to prevent a hyperosmolar tear film from damaging the ocular surface [153,154,155] and for the management of postrefractive-surgery-induced dry eye syndrome [156]. Indeed, there is an expanding pool of clinical data on the efficacy of osmoprotectants, such as erythritol, taurine, trehalose, carboxymethylcellulose, and L-carnitine, to promote exit from the vicious circle of DED physiopathology. Osmoprotectants are small organic molecules that are used in many cell types to restore cell volume and stabilize protein function, allowing adaptation to hyperosmolarity [3]. Among these osmoprotectant molecules, L-carnitine (a natural antioxidant) has also been hypothesized to be crucial by inhibiting some inflammation pathways, such as TRPV1, in the ocular surface [157,158]. Thus, it was shown that L-carnitine osmoprotective effect is elicited through the suppression of hypertonic-induced TRPV1 activation in HCE [159]. Moreover L-carnitine showed a potential to reduce stromal fibrosis through the suppression of TRPV1 activation in human corneal keratocytes during the corneal stromal wound healing process [160].

##### TRPV1 and Ocular Pain

TRPV1 is widely expressed in sensory neurons from trigeminal ganglion [128,161], notably in unmyelinated slowly conducting neurons (C-fibers) and in some corneal Aδ fibers. Once TRPV1 is activated, an increase in AP firing and neuropeptides release, such as calcitonin-gene-related peptide (CGRP) and neurokinins or substance P (SP), occurs in peripheral sensory nerve fibers [162]. As a result, numerous immune and other cell types and proinflammatory mediators are generated, which lead to a positive signaling feedback, causing TRPV1 channel activation and nociceptive signaling. Prolonged TRPV1 activation causes a massive Ca^2+^ influx, which leads to the increased expression of several nociceptive genes.

Capsaicin is an active agent in hot chili peppers [163]. This molecule produces a sharp painful burning sensation. Following this sensation, the mucous membrane that was in contact with capsaicin becomes unresponsive to any noxious stimuli for a long time. In addition, different animal studies have shown using capsaicin injections in neonatal rats that this molecule induces its effect through TRPV1 [164]. This receptor is implicated in different pathological symptoms, such as pain, visceral hyperreflexia, and neurogenic inflammation [165]. Recently, Kishimoto et al. demonstrated that chronic intermittent hypoxia could be responsible for pain on the ocular surface via TRPV1-dependent mechanisms [166].

A lot of studies have shown that TRPV1 is highly implicated in the underlying mechanisms of peripheral sensitization [165,167,168]. TRPV1^−/−^ mice present abnormal nociceptive, inflammatory, and hypothermic responses to vanilloid compounds. Moreover, sensory neurons from such mice do not respond to capsaicin, resiniferatoxin, protons, or temperature (<50 °C). In addition, TRPV1^−/−^ mice do not respond in vivo to capsaicin, and responses to acute thermal stimuli are diminished [125,169]. Moreover, Bereiter et al. showed that excision of the extraorbital lacrimal gland induced chemical corneal hypersensitivity in rats. After ocular instillation of capsaicin or hypertonic saline solution, they evidenced an increase in eye wiping behavior correlated with a greater orbicularis oculi muscle activity in DED animals. In addition, in a immunostaining study, they observed an increase in TRPV1 protein levels in the eye and TG from dry eye rats [104]. In contrast, Yamazaki et al. showed a decrease in TRPV1 in the cornea of DED animals (following excision of the extraorbital lacrimal gland) [170].

Different studies have used preclinical models of DED to better understand the implication of TRPV1 in the increased ocular nociception and pain secondary to DED. Thus, Hatta et al. observed that an application of capsaicin on the cornea induced sustained discharge of ongoing activity at 35 °C for several minutes in DED animals. Moreover, capsaicin decreased corneal cooling responses at 20 °C in DED and control animals. These capsaicin effects were blocked by the application of a TRPV1 antagonist (capsazepine). More recently, after DED induced by lachrymal gland excision (LGE), it was established that corneal axon terminals show a decrease in their density, followed by a regeneration process. After LGE-induced corneal surface alterations, TRPV1 expression is particularly increased in isolectin B4-positive (IB4) TG neurons [171].

Acute ocular exposure to ultraviolet (UV)-B radiation and nonsolar UV-C produces an inflammation of the cornea called photokeratitis. Acosta et al. modelized UV keratitis by irradiating one eye of albino and pigmented guinea pigs with an UV lamp. Then, they stimulated corneal polymodal nociceptors ex vivo using a jet of gas containing 98.5% CO_2_ for 30 s and recorded ciliary nerve firing in intact and UV-irradiated eyes. Thus, CO_2_ stimulation in animals exposed to UV radiation showed an increase in the mean firing response (imp/s) of polymodal nociceptors, a decrease in latency, and no modification of the postdischarge compared with control animals. Besides, nocifensive response to capsaicin was increased in UV-irradiated animals [79].

Hegarty et al. showed that ocular application of capsaicin (0.1%) in rats induced a noxious response but conversely suppressed spontaneous grooming behavior. However, central corneal afferent terminals that are linked to capsaicin-activated trigeminal neurons were shown to not express TRPV1. These observations underline that the expression of the central TRP channel is not necessarily correlated with the type of stimulus transduced by the peripheral nociceptive terminals [172].

##### TRPV1 Agonists to Alleviate Ocular Pain

Moreover, Bates et al. proposed TRPV1 agonists as a treatment for postoperative or postinjury ophthalmic pain. They reasoned based on a publication by Belmonte et al., who observed that topical capsaicin in a cat’s cornea decreased the corneal sensitivity of Aδ polymodal units to chemical and thermal activation [26]. They observed that RTX (an ultrapotent agonist of TRPV1) alone produced a brief but intense noxious response necessitating topical instillations of lidocaine to the cornea. After administration, RTX analgesia did not affect epithelial wound healing and did not cause any histological damage to the cornea. RTX eliminated the response of capsaicin eye wipe response but did not modify the mechanical sensitivity of the cornea [173]. In addition, Por et al. investigated the effects of single and repeated blast exposure on pain and inflammatory mediators in ocular tissues and evidenced an increase in TRPV1, CGRP, SP, and endothelin 1 (ET-1) expression and neutrophil infiltration in the rats’ corneas [174] and TRPV1, ET-1, and glial fibrillary acidic protein (GFAP) protein expression in the TG.

##### TRPV1 Antagonists to Alleviate Ocular Pain

In 2018, Moreno-Montañés et al. presented tivanisiran (formerly SYL1001) as a treatment of DED. They used tivanisiran a small interfering oligonucleotide of RNA (siRNA) designed to silence the human TRPV1. Preclinical efficacity was evaluated in vivo in a capsaicin-induced ocular pain model in rabbits. These animals were treated topically once daily for 4 days with Tivanisiran and showed a comparable palpebral opening ratio to that of capsazepine. Besides, ocular instillation of this drug improved ocular hyperemia and tear quality in humans [175].

Besides, Fakih et al. provided new arguments towards the pharmacological effectiveness of TRPV1 antagonist instillation against DED-induced sensory abnormalities and anxiety. Thus, by in situ hybridization and RT-PCR analyses, they showed that DED triggered an upregulation of TRPV1 mRNA and induced an overexpression of genes involved in neuropathic and inflammatory pain in the ophthalmic branch of the TG. Moreover, they showed that topical instillations of capsazepine (10 μm) twice a day for 2 weeks in a mouse model of severe DED reduced corneal polymodal responsiveness to heat, cold, and acidic stimulation in ex vivo eye preparations. Consistent with these findings, chronic capsazepine instillation inhibited the upregulation of genes involved in neuropathic and inflammatory pain in the TG of DED animals and reduced corneal pain, as well as anxiety-like behaviors associated with severe DED [176] (Figure 3 and Figure 4).

##### TRPV1 and TRPM8 Interaction in DED and Ocular Pain

In an in vitro model of dry eye, 3-iodothyronamine (3T1AM), an endogenous thyroid hormone metabolite, activates TRPM8 at a constant temperature in the human conjunctival epithelial cells (HCE). Moreover 3T1AM reduces the increase in IL-6 observed after TRPV1 activation by capsaicin [177]. Thus, Lucius et al. showed that this phenomenon happens by a crosstalk between TRPV1 and TRPM8. On the one hand, increases in Ca^2+^ influx induced by TRPM8 agonists were all blocked by BCTC, a mixed TRPV1/TRPM8 antagonist. On the other hand, Ca^2+^ transients induced by TRPV1 agonists were suppressed during exposure to TRPM8 agonists in immortalized human corneal epithelial cells [178].

Cold ocular nociception is characterized by an irritative and burning component [179]. Fengxian et al. hypothesized that this burning sensation could be caused by heat channels, such as TRPV1. Thus, they showed that approximately 47% of TRPM8^+^-labeled neurons also showed immunoreactivity for TRPV1 in retrograde-labeled corneal neurons. Moreover, it was observed that DED increased TRPM8 expression in TRPV1-positive neurons. Thus, they established that TRPV1 can enhance TRPM8 responses to cold. Indeed, by using AMG9810 (a potent and selective TRPV1 antagonist), they demonstrated that this treatment is responsible for a decrease in cold response. Thus, they pointed out that the colocalization of TRPV1 and TRPM8 is essential for cold nociception and cold allodynia by enhancing the responsiveness of TRPM8^+^ neurons to cold and facilitating the release of the neuropeptide SP, which is crucial for the communication between TRPM8^+^ neurons and postsynaptic neurons for cold nociception [180]. Hatta et al. evaluated the sensitivity of cool cells to capsaicin in a model of DED in rats. They showed that DED induced alterations in cool cell response properties, including the increased responsiveness to noxious heat and activation by capsaicin. Li et al. enforced the hypothesis of a crosstalk between TRPV1 and TRPM8 and proved that TRPV1 activity and SP release are required for corneal cold nociception [180].

## 4. Conclusions

In conclusion, DED is a growing public health concern that negatively impacts the quality of life of sufferers, including aspects of physical, social, psychological functioning; daily activities; and workplace productivity. As reviewed, there is a close relationship between aberrant TRP expression and disease of the anterior segment of the eye. However, much effort still needs to be undertaken to determine the causal relationships between specific TRP dysfunctions and the pathophysiological disorders associated with the anterior segment diseases, including dry eye and ocular corneal pain. Thus, this review sheds light on the importance of TRP channels, opening a new avenue for repositioning these channels as potential treatment targets for patients suffering from DED and corneal pain. However, it should be kept in mind that some controversies still persist in the literature on the pharmacological use of TRP agonists vs. antagonists against such ocular pathologies.

## Figures and Tables

**Figure 1 pharmaceutics-14-01859-f001:**
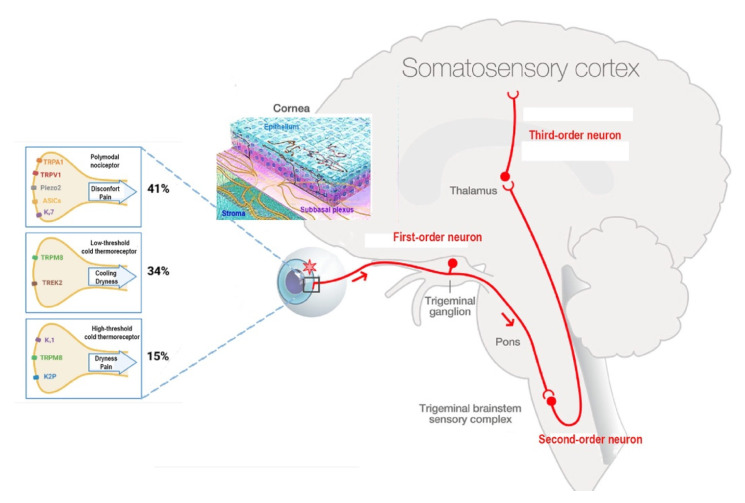
The ocular sensory pathway. “Adapted with the permission from [34] copyright 2022 Elsevier, Melik Parsadaniantz”, first-order neuron: corneal nerve endings present their cell body in the TG and synapse at TBSC levels (red). Second-order neuron connects to contralateral pathways and synapse in the thalamus. At the end, third-order neuron transfers the information to paralimbic region and somatosensory cortex. Corneal epithelium nerve terminals. Corneal nociceptors and cold thermoreceptor and their transducing channels. Percent of corneal mice nociceptors and cold thermoreceptors. Image Adapted with the permission from [32] copyright 2022 IOVS, Melik Parsadaniantz”.

**Figure 2 pharmaceutics-14-01859-f002:**
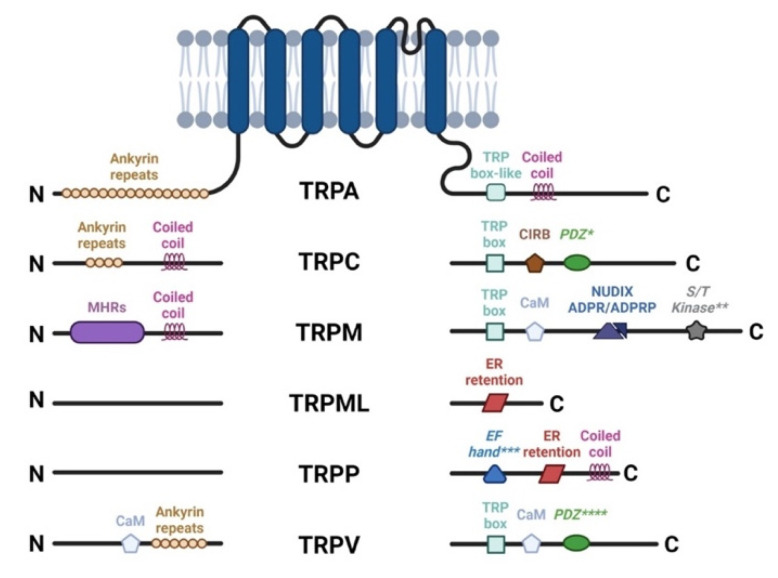
TRP channel architecture. The cytoplasmic N and C termini contain different structural and functional elements, for each subfamily. TRP Box, EWKFAR in TRPCs; AnkR, ankyrin repeats; CC, coiled-coil domain; S/T Kinase, intrinsic serine/threonine kinase (** for TRPM6 and TRPM7); CIRB, calmodulin and inositol triphosphate receptor binding site (InsP3R); PDZ, amino acid motif-binding PDZ domains (* for TRPC4, 5 and **** for TRPV3); NUDIX, NUDT9 homology domain binding ADP ribose or ADPR-2’-phosphate (ADPRP); EF Hand (*** for TRPP1), canonical Ca^2+^ binding domain in TRPP1/PKD2; ER retention, endoplasmic reticulum retention signal.

**Figure 3 pharmaceutics-14-01859-f003:**
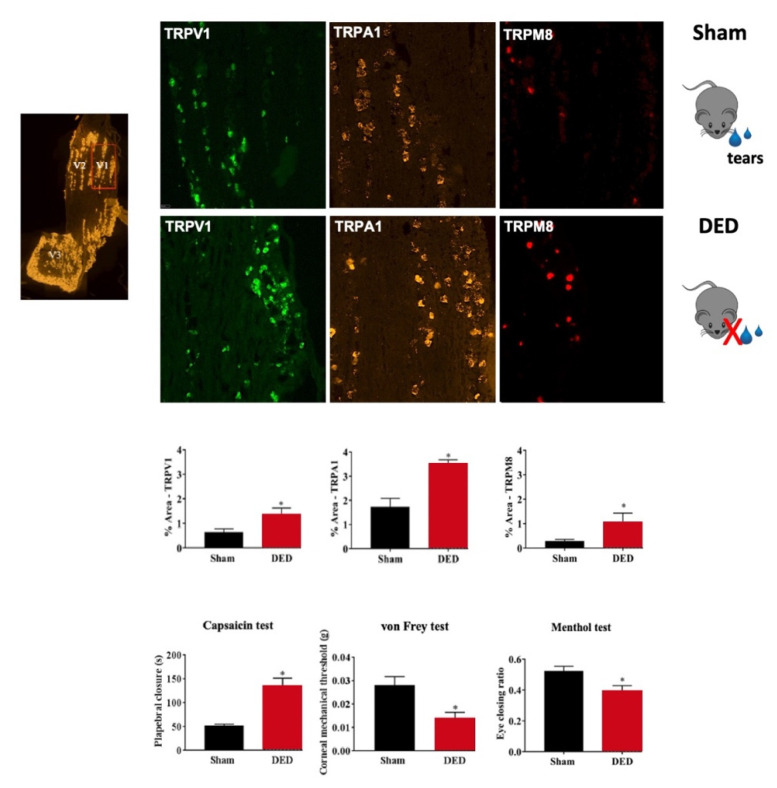
Evaluation of nociceptor expression in the ophthalmic branch of the trigeminal ganglion (TG) of sham and dry eye disease (DED) animals. Localization of the ophthalmic branch (V_1_, red rectangle), maxillary branch (V_2_), and mandibular branch (V_3_) in a mouse TG using the positive probe of RNAScope. Staining and quantification of mRNA levels by in situ hybridization of TRPV1, TRPA1, and TRPM8 in the ophthalmic branch of the TG of sham and DED animals. Evaluation of chemical and mechanical corneal sensitivity of sham and DED animals on d21. Chemical corneal sensitivity was evaluated by using a drop of capsaicin (100 μm) and recording the palpebral closure time for 5 min. The corneal mechanical threshold was measured using von Frey filaments. The cold corneal sensitivity was evaluated by using a drop of menthol (50 µm) and recording the eye closing ratio after 5 min. All experiments were conducted on d21 post lacrimal gland excisions. * *p* < 0.05 relative to the sham group. Results are expressed as the mean ± SEM. Images and bar charts adapted from [105].

**Figure 4 pharmaceutics-14-01859-f004:**
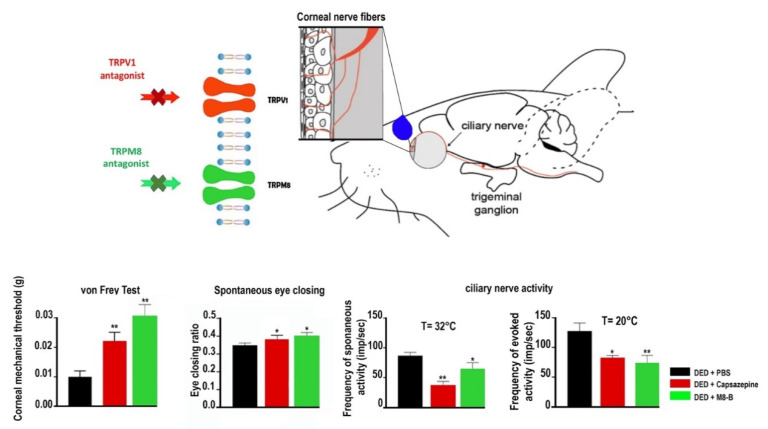
Mechanical allodynia, eye closing ratio of DED animals treated (twice daily from day 7 to day 21 after lacrimal gland excisions) with PBS, 10 μM capsazepine, or 20 µm M8-B. Corneal mechanical sensitivity measured with von Frey filaments. Quantification of the spontaneous eye closing ratio calculated by measuring the width/height ratio. Ex vivo evaluation of spontaneous and evoked activity of the ciliary nerve in DED mice treated (twice daily from day 7 to day 21 after lacrimal gland excisions) with PBS, 10 µM capsazepine, or 20 µm M8-B. Histograms show a mean value of spontaneous activity of the ciliary nerve at 32 °C and evoked activity of the ciliary nerve at 20 °C in DED mice. All experiments were conducted at day 21. * *p* < 0.05 and ** *p* < 0.01 relative to DED mice treated with PBS. Results are expressed as mean ± SEM. Images and bar charts adapted from [105].

**Table 1 pharmaceutics-14-01859-t001:** Functions and localization of TRP channels in the anterior segment of the eye. Respective agonists/stimulators or antagonists/inhibitors of TRP channels.

*TRP Channel Subtype*	Functions	Pharmacological Compounds (Antagonists/Inhibitors/Agonists/Activators)	Anterior Segment Cell-Type Expression
* **TRPC1** *	mechanotransduction	2-APB (2-aminoethoxydiphenylborate)	corneal epithelial cells; trabecular meshwork.
* **TRPC2** *	?	DAG	trabecular meshwork.
* **TRPC3** *	normal mechanotransduction	GSK2332255B; Pyr3; 2-APB; DAG; Pyr3; OAG; GSK1702934A; OptoBI-1.	corneal epithelial cells; trabecular meshwork.
* **TRPC4** *	corneal epithelial cell proliferation	2-APB; GSK3395879; englerin A.	corneal epithelial and endothelial cells.
* **TRPC6** *	cooperative mechanism with TRPV1 in hyperalgesia	GSK2332255B; BI-749327; SAR7334 ; DS88790512.	

* **TRPV1** *	heat sensor > 42 °C; osmotic sensor; low pH; thermal hyperalgesia; neurogenic inflammation; corneal healing and fibrosis.	capsazepine; AMG 517; AMG9810; SB-366791; SB-705498; JNJ-17203212; Asivatrep; Mavatrep; L-R4W2 TFA; V116517; si RNA Tivanisiran; Resolvin D2; WIN55,212-2; A784168; capsaicin; olvanil; mdr-652; resiniferatoxin; anandamide (AEA); Bradykinins; PGE2; ATP; Glutamate, NGF; bisandrographolide C.	corneal nerve fibers; epithelial and endothelial corneal cells; conjunctival cells; corneal keratinocytes.
* **TRPV2** *	heat sensor T > 52 °C.	Probenecid; cannabidiol; 2-APB.	basal layer epithelium of the conjunctiva
* **TRPV3** *	moderate heat sensor T > 30–39 °C; cell viability; corneal wound healing.	2-APB; camphor; carvacrol; farnesyl pyrophosphate; bisandrographolide C.	corneal epithelial cells; corneal endothelial cells; vascular endothelial cells
* **TRPV4** *	osmotic sensor; moderate heat sensor T > 24 °C to 34 °C; mechanosensor; corneal epithelial cell differentiation.	HC-067047; GSK205; GSK2193874; RN-1734; GSK2798745; GSK 1016790A; RN-1747; A. arachidonic; 4-phorbol-12,13- didecanoate; AEA; 2-AG.	corneal epithelial and endothelial cells; conjunctival cells.

* **TRPA1** *	neurogenic inflammation; noxious cold temperature sensor T < 17 °C; mechanical-gated transduction; macrophage infiltration; stromal neovascularization; corneal fibrosis.	HC-030031; A-967079; AM-0902; PF-04745637; GDC-0334; Resolvin D2; AP-18; allyl isothiocyanate; allicin; diallyl disulfide; thiosulfinate; methyl salicylate; formalin; carvacrol; unsaturated aldehydes (acrolein, isocyanate); oxidizing agents; hydrogen peroxyde; PF-4840154; ASP7663; JT010; diphenyleneiodonium chloride; AEA; THC.	corneal epithelial cells; corneal nerve fibers.

* **TRPM1** *	light transduction signal	mGluR6	
* **TRPM2** *	mechanotransduction	H_2_O_2_; ADP-Ribose; *β* NAD+	corneal endothelial cells; trabecular meshwork.
* **TRPM7** *		spermine	
* **TRPM8** *	moderate cold sensor	M8-B hydrochlorid; RN-1747; RQ-00203078; PF-05105679; AMG 333; AMG2850; AMG9678; PF-05105679; AEA; BCTC; THC; menthol; icilin; eucalyptol, WS3; WS-12; D-3263 hydrochloride; FEMA 4809; RN1747.	corneal nerve fibers; corneal epithelial cells; corneal endothelial cells; corneal keratinocytes.

* **TRPP2** *	regulation of tear production	amiloride	basal cell layer of the corneal epithelium; acinar tear cells; lacrimal gland.

## Data Availability

All the reference used in this review are cited in the bibliography section. All the reference are available in pubmed.

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
