# Peer review of "Transient Receptor Potential Channels: Important Players in Ocular Pain and Dry Eye Disease"

_pharmaceutics, 2022, doi:10.3390/pharmaceutics14091859_

Round 1

Reviewer 1 Report

The review by Fakih et al. compiles valuable information on the role of transient receptor potential channels (TRPs) in ocular pain and dry eye disease (DED). The review is of interest for people working on this field and could be suitable for publication in Pharmaceutics. However, there are important points that need to addressed before acceptance.

The implications of some TRP channels is not actualized. For instance, for TRPV1 and TRPM8 there are many publications in recent years (2020-2022) that are not included. This must be completed.

Page 3, L126: Add the following reference, Bon, RS et al. Ann. Rev. Pharmacol. Toxicol. 2021, 62,427.

For section IV.C, add Taiwan J. Ophtalm 202, 10, 106, and mention it containt.

For section IV.F, TRPA1, please comment and add the following references: Int. J. Mol. Sci. 2022, 23, 6629; Front. Pharmacol. 2021, 12, 773871; Mol. Vision 2020, 26, 576.

Section VIII. Conclusions: An indication on the important controversies concerning the use of agonists or antagonists for the same pathology should be added.

There are a number of inconsistencies on typo errors:

Page 4, L157: “…using neonatal capsaicin injections…”, I suppose the authors would indicate “..using capsaicin injections in neonatal animals…”

 L161: “…cells of cental nervous systems” (add “of”)

L167: “..in the underlying…” (“the” instead of “then”)

L2010-211:  Change the sentence, the TRP domain is located at C-terminal in both TRPM and TRPV, and from the sentence this seems the contrary. This is clearly indicated in Fig2

L258-259, L273, L296-298, L406-407, L472,-473: the sentences seem incomplete

L427: downregulation and /or desensitization?

A list of abbreviations could help in a better understanding of the review.

Other minor points:

The way to indicate the references in the text, (1)(2) or (3)(4)(5), should be (1,2) and (3-5), respectively, and similarly all along the text.

References: month of publication are not needed

Author Response

REV #1:

The review by Fakih et al. compiles valuable information on the role of transient receptor potential channels (TRPs) in ocular pain and dry eye disease (DED). The review is of interest for people working on this field and could be suitable for publication in Pharmaceutics. However, there are important points that need to address before acceptance.

First of all, we would like to thank the reviewer for his/her careful reading of our manuscript and for his/her constructive criticisms.

The implications of some TRP channels is not actualized. For instance, for TRPV1 and TRPM8 there are many publications in recent years (2020-2022) that are not included. This must be completed.

We have updated our reference list with the most recent publications as requested.

Page 3, L126: Add the following reference, Bon, RS et al. Ann. Rev. Pharmacol. Toxicol. 2021, 62,427.

This reference has been added to our manuscript; see line 147-148.

For section IV.C, add Taiwan J. Ophtalm 202, 10, 106, and mention it contain.

This reference has also been added to our manuscript and we have mentioned its contents as requested; see lines 318 to 321.

For section IV.F, TRPA1, please comment and add the following references: Int. J. Mol. Sci. 2022, 23, 6629; Front. Pharmacol. 2021, 12, 773871; Mol. Vision 2020, 26, 576.

These references were added to our manuscript, and we commented on them as requested; see line 192 to 193 and 199 to 200.

Section VIII. Conclusions: An indication on the important controversies concerning the use of agonists or antagonists for the same pathology should be added.

We have modified the conclusion accordingly, as follows: ”However, it should be kept in mind that some controversies still persist in the literature between the pharmacological use of TRP agonists vs antagonists against these ocular pathologies. “ lines 506 to 508.

There are a number of inconsistencies on typo errors:

Page 4, L157: “…using neonatal capsaicin injections…”, I suppose the authors would indicate “..using capsaicin injections in neonatal animals…”

 L161: “…cells of cental nervous systems” (add “of”)

L167: “..in the underlying…” (“the” instead of “then”)

L2010-211:  Change the sentence, the TRP domain is located at C-terminal in both TRPM and TRPV, and from the sentence this seems the contrary. This is clearly indicated in Fig2

L258-259, L273, L296-298, L406-407, L472,-473: the sentences seem incomplete

L427: downregulation and /or desensitization?

Thank you for bringing these mistakes to our attention. As requested, all aforementioned typo errors and incomplete sentences were corrected throughout the manuscript.

A list of abbreviations could help in a better understanding of the review.

A list of abbreviations used in the text was added at the end of document in chapter VII.

Other minor points:

The way to indicate the references in the text, (1)(2) or (3)(4)(5), should be (1,2) and (3-5), respectively, and similarly all along the text.

References: month of publication are not needed

Format of references in the text and publication list was corrected accordingly.

Reviewer 2 Report

The review by Fakih and coworkers is widely descriptive, well-written, and encompasses many different aspects concerning the involvement of TRP channels in eye diseases. From my point of view, there are just some minor issues to be corrected before final publication. 

1) In the introduction chapter a wider discussion about the known mechanisms underlying eye dryness should be added. It will help to introduce readers to the use of TRP modulators in the dry eye syndrome.

2) Figures could be added to depict the general function in eye diseases of the different TRP channels described. It would really improve the quality of the review also catching more properly the readers’ attention.

3) Table 1 should be integrated with the references corresponding to the different raws indicated.

Author Response

Rev #2:

The review by Fakih and coworkers is widely descriptive, well-written, and encompasses many different aspects concerning the involvement of TRP channels in eye diseases. From my point of view, there are just some minor issues to be corrected before final publication. 

First of all, we would like to thank the reviewer for his/her careful reading of our manuscript and for his/her fruitful criticisms.

  • In the introduction chapter a wider discussion about the known mechanisms underlying eye dryness should be added. It will help to introduce readers to the use of TRP modulators in the dry eye syndrome.

As requested by reviewers 2 and 3, a chapter describing the mechanisms of eye dryness was added in the introduction from lines 41 to 52.

  • Figures could be added to depict the general function in eye diseases of the different TRP channels described. It would really improve the quality of the review also catching more properly the readers’ attention.

We did not fully understand the reviewer’s request and do not agree that such a figure would be within the scope of our review that focuses specifically on TRP channels in diseases of the anterior segment of the eye. Besides, an excellent and complete review on the involvement of TRP channels in eye diseases was recently published:

Involvement of transient receptor potential channels in ocular diseases: a narrative review

Tian-Jing et al. : Ann Transl Med 2022  https://dx.doi.org/10.21037/atm-21-6145

  • Table 1 should be integrated with the references corresponding to the different raws indicated.

We have chosen not to include references in Table 1 as all references have already been cited in the text, which would make reading this table more cumbersome. In addition, with regard to the column pertaining to pharmacological compounds (antagonists / inhibitors / agonists / activators), all references can be found in pharmacology manuals or supplier sheets.

Reviewer 3 Report

Summary

This well-written comprehensive overview paper listed the current knowledge about the expression, function, and regulation of various TRPs in ocular surface tissues. Specifically, the knowledge of the neurological pathway of the cornea is highlighted since the cornea is the most innervated tissue in the human body. This manuscript is divided into several sections and starts with a detailed introduction of the initial problems with the dry eye disease (DED) such as the pain, which is one of DED’s main symptoms besides foreign body sensation and the like (first section I). Specifically, the review of the current literature starts from the second section (II) focusing of the innervation of the cornea which is provided by peripheral axons of neurons located in the dorsomedial portion of the ophthalmic region of the trigeminal ganglion. A first subsection (A) describes different types of corneal nociceptors and (cold) thermoreceptors. The third section describes and reviews the expression of of transient receptor potential channels (TRPs). At first, there is a general description of TRPs and their (neurological functions). In the subsections A and B, the TRPC- and TRPV families are summarized whereas the latter one is again divided into subsections 1 and 2 (TRPV1 and TRPV2 – 4, respectively). Each subchapter lists the current knowledge of the corresponding TRPs including animal models (e.g. TRPV1 -/- mice). The next chapters C – D describe the TRPM- and the TRPP family. The fourth chapter (IV) specifically introduces to the implication of TRPs in the (patho)-physiology of the anterior segment of the eye, which is also divided into subchapters A – G. Subchapter A briefly summarizes the knowledge of the TRPC4 channel. In contrast, a very extensive review about TRPV1 in different corneal tissues follows in subchapter B including different research approaches (TRPV1 KO mice model). TRPV4, TRPV6, TRPA1 and TRPM8 are reviewed in the following subchapters. The literature searches also include the channel pharmacology of each TRP channel and the effects of drugs (e.g. noxious effects of the TRPA1 channel agonist isocyanates on ocular irritation and pain). In context with DED, the knowledge of the very relevant TRP channel TRPM8 is shown in subchapter G. The following subchapters V – VI describes the knowledge of TRPM8 agonists and antagonists on alleviation of ocular pain whereas chapter VII is focused on TRPV1 and TRPM8 interaction in cold allodynia. The last chapter (VIII) summarizes the conclusions that there is a close relationship between aberrant TRP expression and anterior segment of the eye disease and that much effort still needs to be undertaken to determine the causal relationships between specific TRP dysfunctions and the pathophysiological disorders associated with the anterior segment diseases in connection with DED and pain.

Major comment

On the one hand, this is a very detailed and important contribution for a better understanding of the function of certain TRPs in different ocular tissues of the anterior part of the eye. On the other hand, the structure of this paper with its sections and two subsections may be a little confusing since same TRP channels are reviewed in different (sub)sections. This is quite okay, but perhaps the outline could be improved. Alternatively, the structure from chapter III can be changed as follows: Introduction to the TRPs (as already shown) followed with the knowledge about each TRP subtype family (no further subdivision) and may also include aspects concerning the implication of TRP channels in the (patjho)physiology of anterior segment of the eye. Maybe another reviewer has a suggestion (?).

Another lack is the understanding that the hypertonic-induced TRPV1 activation is an important issue in DED leading to cell shrinkage of the corneal epithelial cells on the eye surface. At this point, different clinical approaches are available to suppress TRPV1-induced calcium influx and connected release of proinflammatory cytokines. More specifically, osmoprotection was suggested as a new therapeutic principle (1) Osmoprotectiva such as L-carnitine and the like influencing TRPV1 channels led to a relieve of patients having DED symptoms (2-9). This manuscript would be strengthened if this aspects would be still considered. It would therefore also be appropriate to create a specific subchapter with the aforementioned clinical approaches that also includes basic studies.

Specific comments

From line 13 (Introduction). In this section, it is suggested to add aspects of hyperosmolarity in connection with DED (10-12). This is important since TRPV1 can be activated by this mechanism, which should also (better) be highlighted in the corresponding chapter concerning the role of TRPV1 in DED.

From line 72: The mechanism of pain trigger could be better explained in an electrophysiological sight. In brief, TRP channel activation (e.g. TRPV1, TRPA1 pain receptors) leads to an influx of cations, which shifts the cell membrane potential to the positive direction (depolarization). This in turn can activate an action potential from a certain membrane voltage threshold.

Line 108: It is recommended to mention that there is a link between the TRPs and GPCRs (GPCR-TRP channel axis) (13) (also from line 151 and 185, 234) (consideration the review by Veldhuis et al).

Line 196: TRPA1 can also be specifically activated by cinnamaldehyde and that there is a difference in pathophysiological patterns in comparison with menthol concerning TRPM8 (14).

Form line 241: A putative DED treatment using the L-carnitine approach can be included in this paragraph (see above) (4, 15, 16). The connection to TRPV1 should be highlighted. Alternatively, these and other (clinical) aspects can be highlighted in a separate chapter only focusing on clinical relevance.

Line 437: To my knowledge, Borneol is only approved in China. In Europe, this drug is probably still not approved because of adverse side effects. This should be clarified an considered in the manuscript if applicable.

Line 234: The inhibitory effect of CB1 agonist (WIN) on TRPV1 activity should be more clearly described and better highlighted.

Table 1: The stroma including corneal keratocytes/fibroblasts also express TRPs such as TRPV1 and TRPM8 (17, 18). It is recommended to complete the table with this additional information.

Minor

Line 88: However, some of these fibers…

Line 151, 185 and 234: Use abbreviation GPCR instead of writing the word in full.

Line 393: Perhaps, there is a typing error (?). Was 5 mM capsazepine really used. It seems very high to me.

References related to this review

1.          Messmer EM. [Osmoprotection as a new therapeutic principle]. Ophthalmologe. 2007;104(11):987-90.

2.          Turan E, Valtink M, Reinach PS, Skupin A, Luo H, Brockmann T, et al. L-carnitine suppresses transient receptor potential vanilloid type 1 activity and myofibroblast transdifferentiation in human corneal keratocytes. Lab Invest. 2021;101(6):680-9.

3.          Seen S, Tong L. Dry eye disease and oxidative stress. Acta Ophthalmol. 2018;96(4):e412-e20.

4.          Hazarbassanov RM, Queiroz-Hazarbassanov NGT, Barros JN, Gomes JAP. Topical Osmoprotectant for the Management of Postrefractive Surgery-Induced Dry Eye Symptoms: A Randomised Controlled Double-Blind Trial. J Ophthalmol. 2018;2018:4324590.

5.          Hua X, Su Z, Deng R, Lin J, Li DQ, Pflugfelder SC. Effects of L-carnitine, erythritol and betaine on pro-inflammatory markers in primary human corneal epithelial cells exposed to hyperosmotic stress. Curr Eye Res. 2015;40(7):657-67.

6.          Hua X, Deng R, Li J, Chi W, Su Z, Lin J, et al. Protective Effects of L-Carnitine Against Oxidative Injury by Hyperosmolarity in Human Corneal Epithelial Cells. Invest Ophthalmol Vis Sci. 2015;56(9):5503-11.

7.          Khajavi N, Reinach PS, Skrzypski M, Lude A, Mergler S. L-carnitine reduces in human conjunctival epithelial cells hypertonic-induced shrinkage through interacting with TRPV1 channels. Cell Physiol Biochem. 2014;34(3):790-803.

8.          Shamsi FA, Chaudhry IA, Boulton ME, Al-Rajhi AA. L-carnitine protects human retinal pigment epithelial cells from oxidative damage. Curr Eye Res. 2007;32(6):575-84.

9.          Kendler BS. Carnitine: an overview of its role in preventive medicine. Prev Med. 1986;15(4):373-90.

10.        Baudouin C, Aragona P, Messmer EM, Tomlinson A, Calonge M, Boboridis KG, et al. Role of hyperosmolarity in the pathogenesis and management of dry eye disease: proceedings of the OCEAN group meeting. Ocul Surf. 2013;11(4):246-58.

11.        Messmer EM, Bulgen M, Kampik A. Hyperosmolarity of the tear film in dry eye syndrome. Dev Ophthalmol. 2010;45:129-38.

12.        Liu H, Begley C, Chen M, Bradley A, Bonanno J, McNamara NA, et al. A link between tear instability and hyperosmolarity in dry eye. Invest Ophthalmol Vis Sci. 2009;50(8):3671-9.

13.        Veldhuis NA, Poole DP, Grace M, McIntyre P, Bunnett NW. The G protein-coupled receptor-transient receptor potential channel axis: molecular insights for targeting disorders of sensation and inflammation. Pharmacol Rev. 2015;67(1):36-73.

14.        Namer B, Seifert F, Handwerker HO, Maihofner C. TRPA1 and TRPM8 activation in humans: effects of cinnamaldehyde and menthol. Neuroreport. 2005;16(9):955-9.

15.        Baudouin C, Cochener B, Pisella PJ, Girard B, Pouliquen P, Cooper H, et al. Randomized, phase III study comparing osmoprotective carboxymethylcellulose with sodium hyaluronate in dry eye disease. Eur J Ophthalmol. 2012;22(5):751-61.

16.        Evangelista M, Koverech A, Messano M, Pescosolido N. Comparison of three lubricant eye drop solutions in dry eye patients. Optom Vis Sci. 2011;88(12):1439-44.

17.        Yang Y, Yang H, Wang Z, Mergler S, Wolosin JM, Reinach PS. Functional TRPV1 expression in human corneal fibroblasts. Exp Eye Res. 2013;107:121-9.

18.        Turker E, Garreis F, Khajavi N, Reinach PS, Joshi P, Brockmann T, et al. Vascular Endothelial Growth Factor (VEGF) Induced Downstream Responses to Transient Receptor Potential Vanilloid 1 (TRPV1) and 3-Iodothyronamine (3-T1AM) in Human Corneal Keratocytes. Front Endocrinol (Lausanne). 2018;9:670.

Author Response

REV #3:

This well-written comprehensive overview paper listed the current knowledge about the expression, function, and regulation of various TRPs in ocular surface tissues. Specifically, the knowledge of the neurological pathway of the cornea is highlighted since the cornea is the most innervated tissue in the human body. This manuscript is divided into several sections and starts with a detailed introduction of the initial problems with the dry eye disease (DED) such as the pain, which is one of DED’s main symptoms besides foreign body sensation and the like (first section I). Specifically, the review of the current literature starts from the second section (II) focusing of the innervation of the cornea which is provided by peripheral axons of neurons located in the dorsomedial portion of the ophthalmic region of the trigeminal ganglion. A first subsection (A) describes different types of corneal nociceptors and (cold) thermoreceptors. The third section describes and reviews the expression of of transient receptor potential channels (TRPs). At first, there is a general description of TRPs and their (neurological functions). In the subsections A and B, the TRPC- and TRPV families are summarized whereas the latter one is again divided into subsections 1 and 2 (TRPV1 and TRPV2 – 4, respectively). Each subchapter lists the current knowledge of the corresponding TRPs including animal models (e.g. TRPV1 -/- mice). The next chapters C – D describe the TRPM- and the TRPP family. The fourth chapter (IV) specifically introduces to the implication of TRPs in the (patho)-physiology of the anterior segment of the eye, which is also divided into subchapters A – G. Subchapter A briefly summarizes the knowledge of the TRPC4 channel. In contrast, a very extensive review about TRPV1 in different corneal tissues follows in subchapter B including different research approaches (TRPV1 KO mice model). TRPV4, TRPV6, TRPA1 and TRPM8 are reviewed in the following subchapters. The literature searches also include the channel pharmacology of each TRP channel and the effects of drugs (e.g. noxious effects of the TRPA1 channel agonist isocyanates on ocular irritation and pain). In context with DED, the knowledge of the very relevant TRP channel TRPM8 is shown in subchapter G. The following subchapters V – VI describes the knowledge of TRPM8 agonists and antagonists on alleviation of ocular pain whereas chapter VII is focused on TRPV1 and TRPM8 interaction in cold allodynia. The last chapter (VIII) summarizes the conclusions that there is a close relationship between aberrant TRP expression and anterior segment of the eye disease and that much effort still needs to be undertaken to determine the causal relationships between specific TRP dysfunctions and the pathophysiological disorders associated with the anterior segment diseases in connection with DED and pain.

We would like to thank the reviewer for his/her careful reading of our manuscript, positive comments, and constructive criticisms.

Major comment

On the one hand, this is a very detailed and important contribution for a better understanding of the function of certain TRPs in different ocular tissues of the anterior part of the eye. On the other hand, the structure of this paper with its sections and two subsections may be a little confusing since same TRP channels are reviewed in different (sub)sections. This is quite okay, but perhaps the outline could be improved. Alternatively, the structure from chapter III can be changed as follows: Introduction to the TRPs (as already shown) followed with the knowledge about each TRP subtype family (no further subdivision) and may also include aspects concerning the implication of TRP channels in the (patjho)physiology of anterior segment of the eye. Maybe another reviewer has a suggestion (?).

We fully agree with the reviewer's comment. Therefore, we have completely reformatted the structure of the text and merged into a single chapter with subchapters all the paragraphs dealing with the same TRP channel family.

Another lack is the understanding that the hypertonic-induced TRPV1 activation is an important issue in DED leading to cell shrinkage of the corneal epithelial cells on the eye surface. At this point, different clinical approaches are available to suppress TRPV1-induced calcium influx and connected release of proinflammatory cytokines. More specifically, osmoprotection was suggested as a new therapeutic principle (1) Osmoprotectiva such as L-carnitine and the like influencing TRPV1 channels led to a relieve of patients having DED symptoms (2-9). This manuscript would be strengthened if this aspects would be still considered. It would therefore also be appropriate to create a specific subchapter with the aforementioned clinical approaches that also includes basic studies.

This issue has been addressed as mentioned in a comment below.

Specific comments

From line 13 (Introduction). In this section, it is suggested to add aspects of hyperosmolarity in connection with DED (10-12). This is important since TRPV1 can be activated by this mechanism, which should also (better) be highlighted in the corresponding chapter concerning the role of TRPV1 in DED.

We added a new paragraph in the introduction section describing the pathophysiology of DED from lines 41 to 52. We also added a new chapter about TRPV1 and DED from lines 390 to 396.

From line 72: The mechanism of pain trigger could be better explained in an electrophysiological sight. In brief, TRP channel activation (e.g. TRPV1, TRPA1 pain receptors) leads to an influx of cations, which shifts the cell membrane potential to the positive direction (depolarization). This in turn can activate an action potential from a certain membrane voltage threshold.

This is now addressed in the text from lines 135 to 136.

Line 108: It is recommended to mention that there is a link between the TRPs and GPCRs (GPCR-TRP channel axis) (13) (also from line 151 and 185, 234) (consideration the review by Veldhuis et al).

We agree and this notion is now developed in the text; see lines 137 to 140.

Line 196: TRPA1 can also be specifically activated by cinnamaldehyde and that there is a difference in pathophysiological patterns in comparison with menthol concerning TRPM8 (14).

Form line 241: A putative DED treatment using the L-carnitine approach can be included in this paragraph (see above) (4, 15, 16). The connection to TRPV1 should be highlighted. Alternatively, these and other (clinical) aspects can be highlighted in a separate chapter only focusing on clinical relevance.

As requested by the reviewer, a new chapter was added in this review describing the role of TRPV1 in DED (lines 389 to 395). Furthermore, we completed this chapter with osmoprotection as a new therapeutic principle against DED; see lines 397 to 407.

Line 437: To my knowledge, Borneol is only approved in China. In Europe, this drug is probably still not approved because of adverse side effects. This should be clarified an considered in the manuscript if applicable.

Thank you for pointing this out. This information is now mentioned in the text; lines 283 to 284.

Line 234: The inhibitory effect of CB1 agonist (WIN) on TRPV1 activity should be more clearly described and better highlighted.

We have rephrased this paragraph to clearly describe the inhibitory activity of CB1 agonist (Win) on TRPV1 activity. See lines 380 to 388.

Table 1: The stroma including corneal keratocytes/fibroblasts also express TRPs such as TRPV1 and TRPM8 (17, 18). It is recommended to complete the table with this additional information.

The table was completed accordingly.

Minor

Line 88: However, some of these fibers…

Line 151, 185 and 234: Use abbreviation GPCR instead of writing the word in full.

The requested corrections have been made.

Line 393: Perhaps, there is a typing error (?). Was 5 mM capsazepine really used. It seems very high to me.

We checked in the publication and confirm that this is indeed the right concentration of capsazepine used in their study.

References related to this review

  1. Messmer EM. [Osmoprotection as a new therapeutic principle]. Ophthalmologe. 2007;104(11):987-90.
  2. Turan E, Valtink M, Reinach PS, Skupin A, Luo H, Brockmann T, et al. L-carnitine suppresses transient receptor potential vanilloid type 1 activity and myofibroblast transdifferentiation in human corneal keratocytes. Lab Invest. 2021;101(6):680-9.
  3. Seen S, Tong L. Dry eye disease and oxidative stress. Acta Ophthalmol. 2018;96(4):e412-e20.
  4. Hazarbassanov RM, Queiroz-Hazarbassanov NGT, Barros JN, Gomes JAP. Topical Osmoprotectant for the Management of Postrefractive Surgery-Induced Dry Eye Symptoms: A Randomised Controlled Double-Blind Trial. J Ophthalmol. 2018;2018:4324590.
  5. Hua X, Su Z, Deng R, Lin J, Li DQ, Pflugfelder SC. Effects of L-carnitine, erythritol and betaine on pro-inflammatory markers in primary human corneal epithelial cells exposed to hyperosmotic stress. Curr Eye Res. 2015;40(7):657-67.
  6. Hua X, Deng R, Li J, Chi W, Su Z, Lin J, et al. Protective Effects of L-Carnitine Against Oxidative Injury by Hyperosmolarity in Human Corneal Epithelial Cells. Invest Ophthalmol Vis Sci. 2015;56(9):5503-11.
  7. Khajavi N, Reinach PS, Skrzypski M, Lude A, Mergler S. L-carnitine reduces in human conjunctival epithelial cells hypertonic-induced shrinkage through interacting with TRPV1 channels. Cell Physiol Biochem. 2014;34(3):790-803.
  8. Shamsi FA, Chaudhry IA, Boulton ME, Al-Rajhi AA. L-carnitine protects human retinal pigment epithelial cells from oxidative damage. Curr Eye Res. 2007;32(6):575-84.
  9. Kendler BS. Carnitine: an overview of its role in preventive medicine. Prev Med. 1986;15(4):373-90.
  10. Baudouin C, Aragona P, Messmer EM, Tomlinson A, Calonge M, Boboridis KG, et al. Role of hyperosmolarity in the pathogenesis and management of dry eye disease: proceedings of the OCEAN group meeting. Ocul Surf. 2013;11(4):246-58.
  11. Messmer EM, Bulgen M, Kampik A. Hyperosmolarity of the tear film in dry eye syndrome. Dev Ophthalmol. 2010;45:129-38.
  12. Liu H, Begley C, Chen M, Bradley A, Bonanno J, McNamara NA, et al. A link between tear instability and hyperosmolarity in dry eye. Invest Ophthalmol Vis Sci. 2009;50(8):3671-9.
  13. Veldhuis NA, Poole DP, Grace M, McIntyre P, Bunnett NW. The G protein-coupled receptor-transient receptor potential channel axis: molecular insights for targeting disorders of sensation and inflammation. Pharmacol Rev. 2015;67(1):36-73.
  14. Namer B, Seifert F, Handwerker HO, Maihofner C. TRPA1 and TRPM8 activation in humans: effects of cinnamaldehyde and menthol. Neuroreport. 2005;16(9):955-9.
  15. Baudouin C, Cochener B, Pisella PJ, Girard B, Pouliquen P, Cooper H, et al. Randomized, phase III study comparing osmoprotective carboxymethylcellulose with sodium hyaluronate in dry eye disease. Eur J Ophthalmol. 2012;22(5):751-61.
  16. Evangelista M, Koverech A, Messano M, Pescosolido N. Comparison of three lubricant eye drop solutions in dry eye patients. Optom Vis Sci. 2011;88(12):1439-44.
  17. Yang Y, Yang H, Wang Z, Mergler S, Wolosin JM, Reinach PS. Functional TRPV1 expression in human corneal fibroblasts. Exp Eye Res. 2013;107:121-9.
  18. Turker E, Garreis F, Khajavi N, Reinach PS, Joshi P, Brockmann T, et al. Vascular Endothelial Growth Factor (VEGF) Induced Downstream Responses to Transient Receptor Potential Vanilloid 1 (TRPV1) and 3-Iodothyronamine (3-T1AM) in Human Corneal Keratocytes. Front Endocrinol (Lausanne). 2018;9:670.

The aforementioned references are now cited in the text.

Reviewer 4 Report

In the proposed review, Fakih and colleagues propose an extensive review of the literature regarding the implication of TRP cation channels on dry eye disease as well as ocular pain.

First at all, this review is impressive and close to be exhaustive.

The authors first present the physiology of corneal innervation with the description of the nociceptive fibers.

In a second part, they presented a general review of TRP channels with reference to the history of their discovery.  Note that such reviews are already available, but that one is clear and thus might be a good reference for lecturer which are not familiar with these channels. It appears that, in their description of the TRP subfamily, the authors leave the larger part to TRPV, in particular TRPV1 which has its own paragraph. While surprising, this could be explained by the specific role of this channel in eye perturbations, as described latter in the review. However this paragraph on TRPV1 may benefit from simplification.

The third part regarding the role of TRP channels in anterior segment of eye is certainly the most original part of the review. It is thus very interesting but sometimes difficult to follow since the authors give extensive description for each single study, in particular for TRPV1. Few sentences to summarize the finding would be helpful. Simplification will be welcome.

The three last chapters regarding the effect of TRPM8 modulation are interesting to demonstrate that positive or negative modulation of the same channel may have therapeutics applications.

The figures are well prepared and correctly complete the text.

Author Response

REV #4:

In the proposed review, Fakih and colleagues propose an extensive review of the literature regarding the implication of TRP cation channels on dry eye disease as well as ocular pain.

First at all, this review is impressive and close to be exhaustive.

We thank the reviewer for his/her kind and constructive comments on our review.

The authors first present the physiology of corneal innervation with the description of the nociceptive fibers.

In a second part, they presented a general review of TRP channels with reference to the history of their discovery.  Note that such reviews are already available, but that one is clear and thus might be a good reference for lecturer which are not familiar with these channels. It appears that, in their description of the TRP subfamily, the authors leave the larger part to TRPV, in particular TRPV1 which has its own paragraph. While surprising, this could be explained by the specific role of this channel in eye perturbations, as described latter in the review. However this paragraph on TRPV1 may benefit from simplification.

As requested by reviewer 3, we have completely reformatted the structure of the text and merged into a single chapter with sub-chapters all paragraphs dealing with the same family of TRP channels. In addition, an effort of simplification was made in the paragraph dealing with TRPV1.

The third part regarding the role of TRP channels in anterior segment of eye is certainly the most original part of the review. It is thus very interesting but sometimes difficult to follow since the authors give extensive description for each single study, in particular for TRPV1. Few sentences to summarize the finding would be helpful. Simplification will be welcome.

We totally agree and have simplified the text as mentioned in the previous comment.

The three last chapters regarding the effect of TRPM8 modulation are interesting to demonstrate that positive or negative modulation of the same channel may have therapeutics applications.

The figures are well prepared and correctly complete the text.

Thank you for your positive comments.

Round 2

Reviewer 1 Report

The new version is more complete, but still lacks a couple of previous recent reviews on TRPM8: Int. J. Mol. Sci. 2021, 22, 8502;  and Eur. J. Pharmacol. 2020, 882,173321, and some comments on two recent patents: WO2022150714 and WO2020178429, claiming on the use of TRPM8 agonists for DED. 

A few minor corrections: 

L261,453: Agonists (use the plural)

L285,464 Antagonists (as above)

Author Response

The new version is more complete, but still lacks a couple of previous recent reviews on TRPM8: Int. J. Mol. Sci. 2021, 22, 8502;  and Eur. J. Pharmacol. 2020, 882,173321, and some comments on two recent patents: WO2022150714 and WO2020178429 and, claiming on the use of TRPM8 agonists for DED. 

We searched in Google patents for the two patents cited in reference by the reviewer #1 and we think that one is outside the scope of the current paragraph

WO2022150714 : the invention relates to Top-mounted air conditioner used in enclosure cooling ??

WO2020178429: the invention relates to an ophthalmic composition comprising a nonionic surfactant as active ingredient, wherein the non-ionic surfactant is present in an effective amount from 0.01 to 8 weight/volume % based on the total composition, for use in treating or preventing dry eye and/or for relieving one or more symptoms and/or signs of dry eye

As requested by the referee, we add the following sentence Lines 260 to 262 Recently, C. Izquierdo et al. reviewed in depth, the different structural studies of the TRPM8 channel as well as its pharmacological modulation by specific agonists/antagonists.

And

Lines 283 to 284: Additionally, some patents report the use of TRPM8 agonists to relieve ocular pain and dry eye.

Concerning the second reference cited by the referee #1, unfortunately this reference is wrong and consequently is out of the scope of this review.

Eur J Pharmacol . 2020 Sep 5;882:173321. doi: 10.1016/j.ejphar.2020.173321. Epub 2020 Jun 29. :The effect of O-1602, a GPR55 agonist, on the cyclophosphamide-induced rat hemorrhagic cystitis

A few minor corrections: 

L261,453: Agonists (use the plural)

L285,464 Antagonists (as above)

We use plural as demanded (Lines 261, 453 and 285,464).